# Inland Waterway Infrastructure Maintenance Prediction Model Based on Network-Level Assessment

Fan Zhang [1,2], Pingyi Wang [1,*], Huaihan Liu [3], Bin Zhang [1] , Jianle Sun [4] and Jian Li [1]

1   Key Laboratory of Hydraulic and Waterway Engineering of Ministry of Education, Chongqing Jiaotong University, Chongqing 400074, China; zf9214@163.com or zff@yznu.edu.cn (F.Z.); zbin196@163.com (B.Z.); linsanity0920@163.com (J.L.)
2   School of Civil Architectural and Engineering, Yangtze Normal University, Chongqing 408100, China
3   Changjiang Waterway Bureau, Wuhan 430010, China; huaihan_liu@163.com
4   Information Center, Yangtze Normal University, Chongqing 408100, China; jl_sun@yznu.edu.cn
*   Correspondence: py-wang@163.com

**Abstract:** Maintenance decision optimization based on network-level assessment has a long history in road transportation infrastructure and has greatly assisted management departments in saving in expenditure on maintenance costs. However, its application and research in water transportation infrastructure have been lacking. This paper aims to design a predictive model for waterway improvement building maintenance based on network-level assessment and provide a new solution for optimizing the allocation of limited maintenance funds for inland waterway infrastructure. The proposed network-level assessment framework and predictive model comprise data collection, maintenance prediction, and maintenance decision modules. A small time-series dataset was constructed based on the classification proportions of improvement building technical conditions in the jurisdiction of the Yangtze River trunk waterway over the past five years. The two-parameter moving average method was transformed into a single-parameter "jurisdiction moving average method" to suit the characteristics of the dataset. Three models, namely the jurisdiction moving average (JMA), the linear regression (LR), and the quadratic curve regression (QCR) models, were employed to perform calculations on the dataset, which were evaluated using *t*-tests and error analysis. The research results indicated that both the JMA and LR models showed good overall performance and were recommended for use. Especially, the confidence intervals of the JMA model increased the credibility of the prediction results, making it the ideal choice. This study also found that the inland waterway maintenance prediction technology based on the network-level evaluation has higher overall efficiency than the known existing technologies. The proposed predictive model allows for a simple and rapid assessment of the overall risk status of regional waterway facilities and is easy to promote and apply.

**Keywords:** inland waterway; improvement building; network-level assessment; prediction; moving average method; regression analysis; maintenance decision optimization

## 1. Introduction

Water transportation, compared to other modes of transportation, is environmentally friendly and pollution-free, which is conducive to the sustainable development of society and has played an essential role in economic construction from ancient civilizations to the present day [1]. Waterway improvement buildings are important infrastructure for water transportation; they are mainly used to improve and maintain the passage conditions of waterways and enhance waterway traffic capacity. However, due to water erosion, riverbed deformation, material aging, and human-related factors, the improvement structures continuously sustain damage during their service life, resulting in performance degradation and navigation condition deterioration [2,3]. Therefore, the technical condition assessment and repair of improvement buildings are important tasks for waterway maintenance [4,5].

Currently, the maintenance and management of waterway improvement buildings require vast funds, but cyclical operations have low efficiency. According to statistics, investment in Yangtze River trunk waterway improvement building projects, including completed, yet-to-be-completed, and presently underway projects, has exceeded CNY 40 billion over the past 30 years. In the meantime, improvement building maintenance has long work cycles, lasting up to a year in duration, and involves project tendering and bidding, evaluation, maintenance planning, funding, and implementation [4]. Around the world, the economic resources allocated to transportation infrastructure maintenance are limited and often far from sufficient for fully maintaining the required reliability and service levels throughout the life cycle. Therefore, the optimal allocation of existing funds has become the most important goal [6]. With the rapid development of the Yangtze River waterway construction, the comprehensive benefits of the improved buildings are becoming increasingly prominent. This requires rational resource allocation and the unified coordination and operation of waterway management [7]. Assessing the technical condition and maintenance of these buildings is a challenging task. One of the key challenges faced here is the limited efficiency of the current periodic operational model, which further faces constraints due to limited economic resources allocation. Consequently, innovative methods are required to simplify maintenance operations and resource utilization. Additionally, current methods mainly focus on evaluating the current service status of individual building, often overlooking the critical aspect of future overall risks. Therefore, a paradigm shift is needed in the maintenance of harbor remediation buildings, moving from conventional maintenance practices to forward-looking and systematic approaches. Evaluation methods should not only be focused on current conditions but also on predicting future risks.

Researchers have already begun researching to assess the structural stability and performance of waterway improvement buildings. They have adopted various methods to assess factors such as hydraulic elements, scour pit parameters, damage volume ratios, and scour depths, with the aim of improving the safety and maintenance efficiency of these structures.

Scholars worldwide have long studied the structural stability evaluation of waterway improvement buildings. Using the back wall slopes of local erosion pits as the main index for spur dike erosion stability evaluation, Wang et al. [8] established a reliability function to investigate the combined sensitivity of hydraulic elements, scour pit shape, and scour pit size on the index based on measured data with random parameters. Han et al. [9] used the ratio of damage volume as a quantitative index to assess the safety of improvement buildings and the effectiveness of improvement. Yu et al. [10] used the scour depth at the jetty head as a spur dike structural safety evaluation index and comparatively analyzed the extent to which different design parameters influence structural safety using experimental data. Scholars have also studied the performance of improvement buildings. Chen et al. [11] used factors such as the amount and variation of incoming water and sand, water surface slope, riverbed morphology, and riverbed sand composition as evaluation indexes for the function of clearing sands with converging flow in spur dikes and employed the regression support vector machine theory to evaluate and predict the indexes. Muto et al. [12] and Clark et al. [13] included environmental impacts, economic impacts, and other factors in the evaluation indexes to analyze and predict the performance of dams. As can be seen, previous research focused mainly on evaluating a single aspect of improvement buildings. In practice, waterway improvement building service status maintenance is a multi-criteria decision-making (MCDM) problem. Accordingly, Jiang et al. [2], Zhang et al. [14], Wen et al. [15], Li et al. [16], and Wang et al. [17] have also established multi-factor evaluation index sets in recent years. Based on aspects such as appearance deformation, damage degree, and functional operation of waterway improvement buildings, their service status was comprehensively evaluated through different theoretical approaches. However, these methods judge the current service status of improvement buildings and rarely predict their future risk status. While scholars have researched the temporal variations in the functional status of buildings, their focus was primarily on eval-

uating and predicting individual buildings in the lower reaches of the Yangtze River [11], neglecting the distribution of building clusters and the overall risk status. Moreover, existing evaluation methods are highly intricate and demand extensive field data, including building deformation, damage severity, riverbed topography, and hydrological data. In addition, stress analysis needs to be conducted using hydraulic elements such as flow velocity, water level, flow rate, and sediment. Often, the overly complex operational procedures and the difficulties in obtaining many indicator data mean that it is challenging to promote and apply such evaluation methods on a large scale in practical engineering.

Traditional maintenance methods for waterway improvement buildings have limitations, as they operate in a reactive and periodic manner which makes it difficult to efficiently utilize resources and improve operational efficiency. Current evaluation models are mainly focused on individual and passive assessments, evaluating only after issues have already occurred rather than proactively predicting future risks. This limits the effective maintenance of waterway transportation infrastructure and hinders optimal allocation of limited maintenance funding. Although research has advanced our understanding of various aspects of improvement building evaluation, there remains a pressing need for a transition towards more comprehensive and forward-looking maintenance methods.

Research on road transportation infrastructure has divided the evaluation of bridge technical condition indicators into two categories: current assessment and condition development prediction. The former reflects the current structural conditions and mainly focuses on individual buildings, while the latter predicts future conditions (i.e., time-dependent conditions) and is mainly used for the overall assessment and maintenance decision optimization of bridges at the network level [18], known as network-level assessment. Infrastructure management system experiences from the United States have shown that adopting a systematic approach to facility management can bring tremendous benefits in practice. For example, to minimize the life cycle cost of the road surface portion in the network, Arizona made maintenance and recovery resource allocation decisions through a project management system, which saved over USD 200 million in maintenance and repair costs over five years [19]. Network-level assessment is the most commonly used and effective way to comprehensively evaluate and predict bridge network states from a systemic management perspective and is a crucial component of the bridge management system [6,20]. Martinez et al. [21] conducted a network-level assessment of 2802 bridges in Ontario, Canada, based on various bridge condition prediction models, including k-nearest neighbors, decision trees, linear regression, artificial neural networks, and deep learning neural networks. By comprehensively comparing the prediction accuracy, performance, and certainty, the decision tree model was recommended. Adarkwa et al. [22] explored and predicted the network-level performance of bridges in different states of the United States, focusing on the tensor decomposition data analysis method. They defined network-level performance indicators as the percentage of bridges with structural defects, functional obsolescence, or both in the network, calculated based on bridge area and quantity. Xia et al. [23] proposed a comprehensive data-driven framework for network-level bridge condition assessment, incorporating data integration, condition assessment, and maintenance management and applied it to an actual highway bridge network in Hebei Province, China; this revealed the bridge's condition and level of deterioration and the influence of maintenance actions over time through periodic bridge inspection reports obtained from thousands of bridges over multiple years. The results indicated that the proposed data-driven approach could guide bridge managers in estimating future conditions and allocating maintenance resources. Therefore, the network-level assessment approach for bridge system management can also be applied to maintenance decision making for buildings along the Yangtze River waterway improvement project.

In the field of transportation infrastructure, effective maintenance is crucial, as it not only relates to safety and reliability but also to the rational allocation of resources. Although effective maintenance decision models already exist in road transportation, research in this area is relatively scarce relative to waterway transportation. This has motivated

our research to explore new methods for enhancing maintenance efficiency, optimizing resource allocation, and fostering the sustainable development of inland waterway transportation. This research aims to bridge existing research gaps and provide improved tools for management authorities to address maintenance challenges in inland waterway transportation infrastructure.

Maintenance decision optimization methods based on network-level assessment have long been used in the field of road transportation infrastructure, providing significant help to management authorities in saving maintenance costs. Therefore, maintenance decision making for the improvement buildings along the Yangtze River can also draw inspiration from the principles of bridge system management and adopt network-level assessment. The primary purpose of this study is to apply network-level assessment to waterborne transportation infrastructures, collect and organize historical inspection data of multiple waterway improvement buildings, propose a network-level assessment and maintenance prediction method for inland waterway improvement buildings from a system management perspective, and provide a new approach for optimizing the maintenance fund allocation and comprehensive navigational infrastructure management. The main research questions include the following: (1) How can we adjust and apply network-level assessment methods to the maintenance of waterway transportation infrastructure? (2) Is it feasible to use historical inspection data to predict the future risk status of channel improvement buildings?

Given this, the Yangtze River trunk waterway improvement building technical condition categories in recent years were recorded. A network-level evaluation framework with data collection, maintenance prediction, and maintenance decision-making modules was proposed. Through the integration of methods, including moving averages and regression analysis, we calculated the overall technical conditions of improvement buildings in each jurisdictional area, conducted an error test, and proposed a maintenance prediction model for inland waterway improvement building maintenance. The research results can facilitate fast prediction of the overall risk status of regional waterway improvement buildings and provide a scientific basis for the in-advance formulation of waterway improvement building maintenance budgets, thus improving the efficiency and fineness of waterway operation and management and enhancing resource allocation rationality.

## 2. Research Background and Dataset

The research background of the paper is the trunk waterway of the Yangtze River, and the dataset consists of the technical condition evaluation results of the waterway improvement structures collected over the past 5 years.

### 2.1. Research Areas and Objects

With a distant source, a long stream, and abundant ice-free water, the Yangtze River has superior water transportation conditions, earning it the fame granted by the following name: "Golden Waterway". The Yangtze River trunk waterway starts from Shuifu in Yunnan Province and ends at the estuary, which is divided into the upper, middle, and lower reaches, as well as the estuary, with a total length of 2843 km. Of the waterway, 97% is maintained and managed by the Changjiang Waterway Bureau, which has divided it into several jurisdictional sections according to different geographical locations and environmental characteristics (Figure 1 and Table 1).

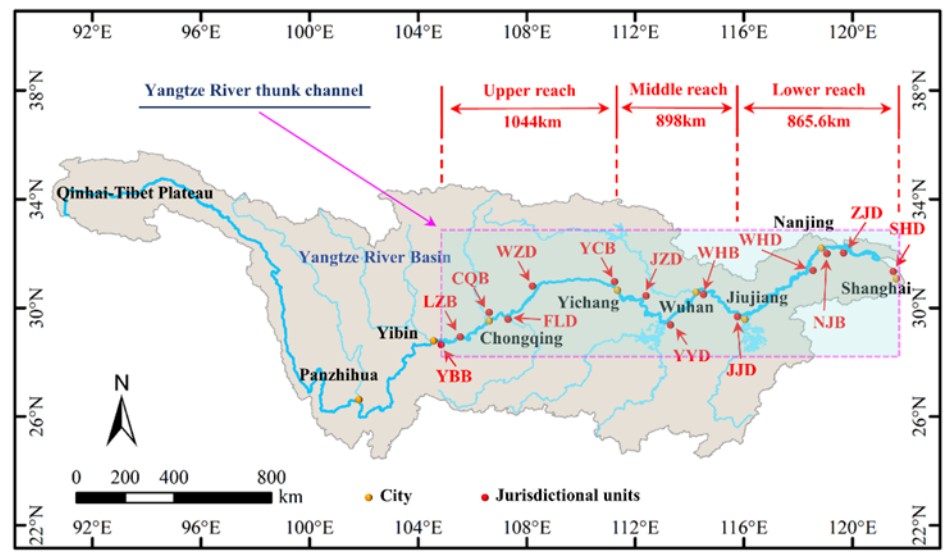

**Figure 1.** Schematic diagram of Yangtze River trunk waterway and jurisdiction sections. The acronyms in red represent the jurisdictional units, the corresponding full names can be found in Table 1; the red dashed line represents the boundary between the upper, middle, and lower reaches of the Yangtze River.

**Table 1.** Yangtze River trunk waterway jurisdiction sections.

| Jurisdictional Units | Abbreviation | Maintenance Range (km) | Yangtze River Reach |
|---|---|---|---|
| Changjiang Yibin waterway bureau | YBB | 91.0 | Upper |
| Changjiang Luzhou waterway bureau | LZB | 210.8 | Upper |
| Changjiang Chongqing waterway bureau | CQB | 130.4 | Upper |
| Changjiang Fuling waterway division | FLD | 203.0 | Upper |
| Changjiang Wanzhou waterway division | WZD | 265.0 | Upper |
| Changjiang Yichang waterway bureau | YCB | 196.0 | Upper and Middle |
| Changjiang Jingzhou waterway division | JZD | 189.0 | Middle |
| Changjiang Yueyang waterway division | YYD | 177.0 | Middle |
| Changjiang Wuhan waterway division | WHB | 349.2 | Middle and Lower |
| Changjiang Jiujiang waterway division | JJD | 128.0 | Lower |
| Changjiang Wuhu waterway division | WHD | 325.3 | Lower |
| Changjiang Nanjing waterway bureau | NJB | 97.7 | Lower |
| Changjiang Zhenjiang waterway division | ZJD | 136.6 | Lower |
| Changjiang Shanghai waterway division | SHD | 131.0 | Lower |

To clear the rapids and maintain the waterway grade, many improvement buildings have been built in the upper, middle, and lower reaches of the Yangtze River. As of 2021, 559 improvement buildings have been completed and put into service, which are classified into three types according to their functions and shape characteristics: dam-type buildings, riverbed-protecting buildings, and bank-slope-protecting buildings. Primarily, these buildings are employed to modify the riverbed morphology and sediment transport, facilitating the creation of favorable hydraulic structures. They harness the energy of water flow for channel scouring and ensure the stability of the channel, thus guaranteeing adequate navigational dimensions, especially during low-water periods. Common improvement buildings include spur dikes, longitudinal dikes, submerged dikes, lock dikes, and X-type flexible mattresses. For the convenience of maintenance, the Changjiang Waterway Bureau evaluates the technical condition of the improvement buildings during the dry season each year. According to the Technical Code of Inland Waterway Maintenance [24] and other relevant standards, the technical condition of waterway improvement buildings is divided into five categories, including the first, second, third, fourth, and "unevaluated" categories; then, appropriate maintenance decision recommendations are offered for each

category to guide waterway infrastructure maintenance [14,17]. For instance, structures with significant damage that negatively impacts their improvement functions may be categorized as the third or fourth categories, leading the decision-making process for repairs by management authorities. The specific categorization criteria are shown in Table 2.

**Table 2.** Inland waterway improvement building technical condition categorization and maintenance standards.

| Building Categories | Technical Condition of the Waterway Improvement Buildings | Recommendations for Maintenance Decisions |
|---|---|---|
| The first | Good technical condition and normal function. | Do not repair |
| The second | The building has a small amount of deformation, but the building stability and improvement function are not affected. | Postpone repair |
| The third | The building has more obvious damage and still has an improvement function, but timely repairs are needed. | Repair |
| The fourth | The building is seriously damaged or has obvious defects and has lost or will lose its improvement function. | Repair |
| Unevaluated | The building is under repair, under renovation, or being commissioned within the jurisdiction. | |

Notes: buildings that are flooded, covered, or have fulfilled their designed functions no longer require maintenance and, therefore, are not involved in evaluation and categorization.

According to Bocchini (2011) [20], bridge maintenance strategies can be categorized into preventive maintenance (PM), essential maintenance (EM), and required maintenance (RM). A comparison of the information suggests that the maintenance strategy adopted by Changjiang Waterway Bureau for repairing individual buildings conforms to the EM strategy, as depicted in Figure 2. Resolutely implementing this strategy can effectively prevent the occurrence of the fourth category's technical conditions.

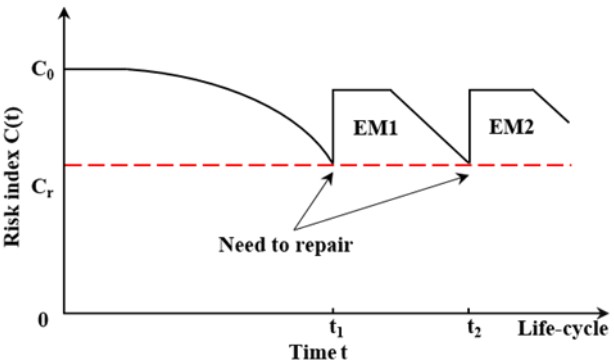

**Figure 2.** Essential maintenance (EM) schematic diagram (C(t) represents a certain risk index).

*2.2. Data Set*

The technical condition evaluation results of nearly 500 improvement buildings along the Yangtze River trunk waterway over the past five years have been gathered, sorted, and compiled into a dataset for further analysis and research in this article. The dataset includes 11 jurisdictional units except for FLD, WZB, and NJB, with evaluation results covering 2017–2022 and classified into the first, second, third, fourth, and unevaluated categories. No building was classified into the fourth category over the past 5 years. FLD, WZB, and NJB construction buildings required no maintenance due to water depth and geographical reasons. The evaluation results for the upper and middle reaches of the Yangtze River in 2017 were missing and not included in the statistics. Figures 3 and 4, respectively, display the technical condition category based on jurisdictional regions and over different time periods.

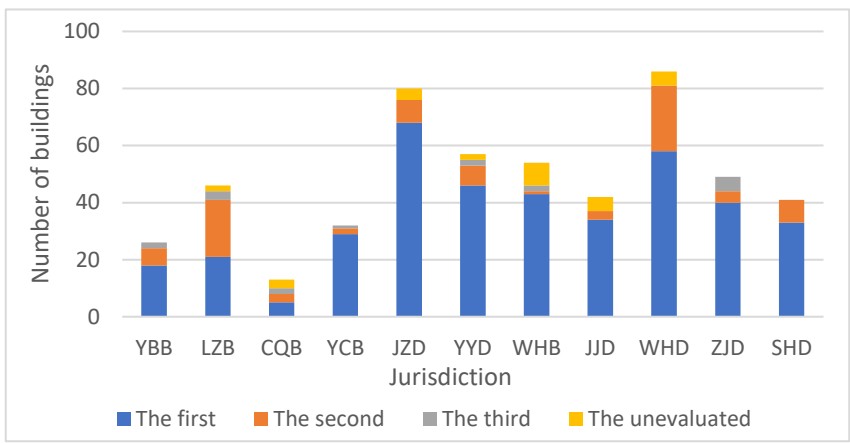

**Figure 3.** Technical condition categories of the Yangtze River trunk waterway according to jurisdiction area statistics (2021).

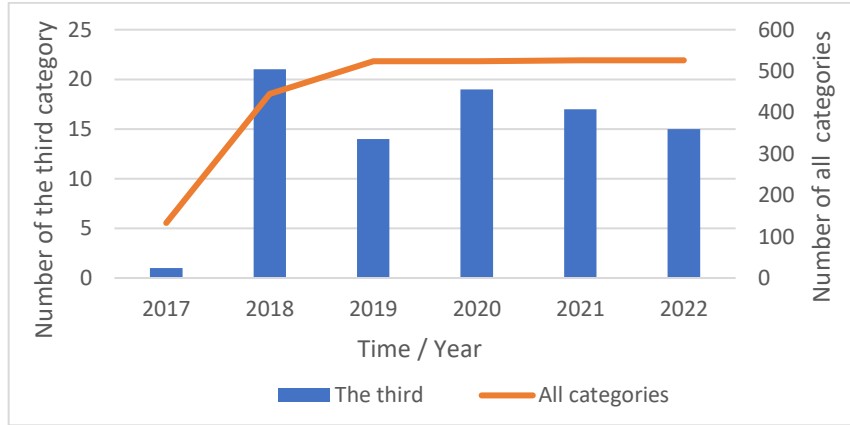

**Figure 4.** Technical condition categories of the Yangtze River trunk channel according to yearly statistics (total for the upper, middle, and lower reaches).

Buildings evaluated as the third category or below require repair, incurring maintenance costs. Thus, their predicted results are more instructive for the management. Using the network-level assessment of the third category buildings as an example, the calculation and discussion of the maintenance prediction model are conducted.

## 3. Methods

### 3.1. Network-Level Assessment Framework

The infrastructure structure and performance within the region have certain correlations and continuities. Specifically, buildings under the same jurisdiction tend to show similar degradation patterns in similar environmental factors (such as hydrological environment, geographical location, navigational guarantee, etc.). In addition, the design specifications, construction standards, and quality of buildings in the same region also tend to converge [25]. Based on this, this paper proposes a network-level assessment framework shown in Figure 5, which consists of data collection, maintenance prediction, and maintenance decision modules.

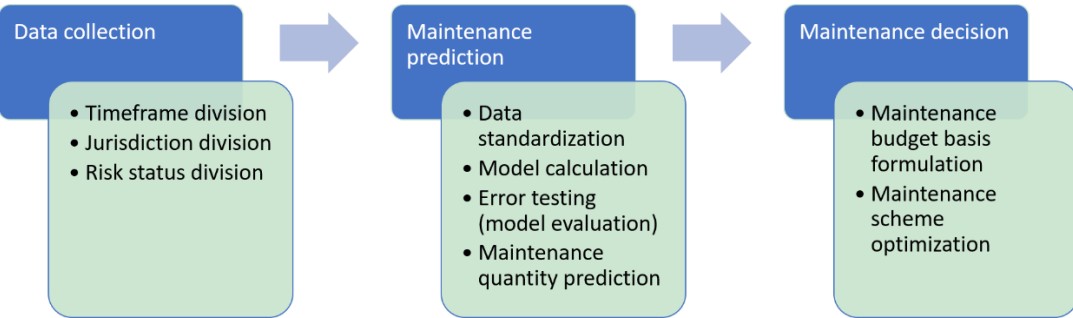

**Figure 5.** Network-level assessment and maintenance decision optimization framework for inland waterway improvement buildings.

The data collection module is mainly responsible for filtering and organizing the historical condition categories of improvement buildings within the jurisdiction area to form a reasonable and effective dataset. The dataset includes three attributes: jurisdiction area, time, and risk level. These are detailed in Section 2.2.

The maintenance prediction module primarily performs data standardization, model calculation, and error testing with the purpose of predicting future maintenance quantities for regional buildings. Were there multiple models to choose from, the module would comprehensively evaluate the model performance and test results to select the best model.

The maintenance decision module converts the final prediction results into the basis for maintenance budgeting and optimizes the overall allocation plan for future funds, thus assisting the management department in maintenance decision making.

### 3.2. Data Standardization

The main purpose of data standardization is to eliminate the differences in quantity between different time periods and regions, making it easier to fit uniformly. The technical condition categorization parameters of the buildings are represented by "category proportion", and the calculation formula is as follows:

$$\text{Category proportion = number of statistical samples of this type/total number of statistical samples} \times 100\% \quad (1)$$

Two points should be noted here: First, the number of "unevaluated" buildings is artificially controlled, mainly related to the maintenance and construction progress, and has nothing to do with the natural state, which will cause interference with the statistical results. Therefore, it should be excluded. Second, the standardized dataset must be divided into the calculation and testing sets. The time-series sample data from 2017 to 2021 are used for model calculation (or fitting) and performance evaluation. The data in 2022 are used for model testing to verify the accuracy and generalization ability of the model.

To ensure prediction accuracy, the range division of the prediction sample set should be as small as possible. Meanwhile, the number of samples per sample set should be over 25–50 to ensure statistical reliability. According to the statistics in Table 1, the number of buildings meets the requirements for all jurisdictions except CQB. In practice, the maintenance of improvement buildings is carried out on a jurisdiction basis. Therefore, the sample set was divided according to jurisdictions to provide statistics and statistically standardizes on the technical condition categories of improvement buildings in each jurisdiction.

### 3.3. Prediction Models

The dataset contains small temporal samples covering only 4–5 historical years of improvement building category data within the jurisdiction. Considering the data characteristics and application scenarios, complex machine learning methods such as neural networks, genetic algorithms, and support vector machines do not apply, while simpler predictive methods such as moving average analysis and regression analysis are more appropriate. According to an in-depth analysis and complexity assessment of the problem,

this study identifies three potential prediction models: moving average, linear regression, and quadratic regression. The next step involves testing the predictive performance of these three models based on actual data and determining which model to employ for practical forecasting purposes.

(1)  Jurisdiction moving average (JMA)

This method is derived from moving average analysis, a classical time-series analysis method that calculates the average value of data over a certain time period by gradually moving through the temporal domain. It can better reflect the trends and changes in the time-series [26]. The construction planning of the Yangtze River waterway is divided by decades, and the jurisdictional waterway status can be assumed to be relatively stable within each decade and will not undergo significant changes. Therefore, the moving average method is suitable for analyzing and predicting the building status.

First, the moving average method must be adjusted based on the data characteristics. For the dataset classifying the technical conditions of buildings in the Yangtze River trunk waterway, geographical parameters should be included in addition to the temporal parameters to form a two-parameter moving average method for network-level assessment and prediction. Furthermore, the excessively small number of sequential samples in the dataset necessitates further temporal division. Therefore, it can be simplified into a single-parameter moving average method, i.e., the jurisdiction moving average method.

The specific procedure is as follows: first, conduct a statistical division of jurisdictional areas (already completed during data standardization, with a moving window size of approximately 25–30 samples measured per unit of building sample); then, gradually calculate the time-series sample averages for each jurisdiction. The calculation formula is as follows:

$$y = \left. \sum_{i=1}^{n} x_i \middle/ n \right.$$

(2)

where $y$ represents the dependent variable of the predictive model, i.e., the proportion of buildings requiring maintenance within a jurisdiction in the upcoming year, $x_i$ represents the time-series sample for a specific jurisdiction, and $n$ represents the size of the time-series sample.

To calculate the possible range of fluctuations in the predicted results, confidence intervals are constructed using t-distribution critical values [27]:

$$\overline{x}_i \pm t_{\alpha/2} \cdot \left. s \middle/ \sqrt{n} \right.$$

(3)

where $\overline{x}_i$ is the sample mean, $s$ is the sample standard deviation, $n$ is the sample size, and $t_{\alpha/2}$ represents the two-tailed critical value associated with the confidence interval and freedom degrees, which can be found in the t-distribution table. Here, $\alpha$ is set to 0.05, the confidence interval is 95%, and the freedom degrees are $n - 1$.

The standard deviation of the time-series sample set is also the root mean square error (RMSE) of the calculation results, and the calculation formula is shown in Equation (6).

(2)  Linear regression (LR)

Linear regression is commonly used to identify the linear relationship between the dependent and independent variables. The fundamental idea is establishing a linear equation to predict the dependent variable while minimizing prediction errors [28]. The equation takes the following form:

$$y = \alpha + \beta x$$

(4)

where y is the dependent variable, x is the independent variable, and $\alpha$ and $\beta$ are constants representing the intercept and slope, which are obtained through the least squares method [27] based on historical data.

(3)   Quadratic curve regression (QCR)

When the time-series sample exhibits significant fluctuations, leading to a large sample standard deviation and, consequently, larger expected deviations, a simple linear fit may not effectively describe the data. In such cases, introducing a quadratic term into the model can better fit the data and capture the curved relationship, which is known as quadratic curve regression [28]. The equation takes the following form:

$$y = ax^2 + bx - c \tag{5}$$

where a, b, and c represent the coefficients of the quadratic, linear, and constant terms, respectively. These coefficients are determined through the least squares method based on historical data.

After obtaining the fitting parameters, future-year proportions for a specific type of building are calculated using Formulas (3)–(5).

*3.4. Error Test*

After model calculation, it is essential to evaluate its performance and conduct model testing to recommend the final predictive model. In this study, a comprehensive comparison of three models is conducted through error tests, where the calculation error and final result error are considered. The considered error parameters include the root mean square error (RMSE) and the coefficient of determination ($R^2$) for the calculation results, as well as the mean absolute error (MAE), mean square error (MSE), and significance level (p-value) for the test results.

RMSE and $R^2$ are used to assess the model's explanatory power. A smaller RMSE value indicates a better fit or calculation performance. $R^2$ ranges between 0 and 1, with values closer to 1 indicating better explanatory power of the model. MAE and MSE, on the other hand, are used to evaluate the model's prediction accuracy and generalization ability. Smaller values for both metrics indicate better predictive performance. The formulas are as follows:

$$\text{RMSE} = \sqrt{\sum_{i=1}^{n} (y_i - \hat{y}_i)^2 / n} \tag{6}$$

$$R^2 = 1 - \left[ \sum_{i=1}^{n} (y_i - \hat{y}_i)^2 / \sum_{i=1}^{n} (y_i - \overline{y}_i)^2 \right] \tag{7}$$

$$\text{MAE} = \sum_{j=1}^{m} |y_j - \hat{y}_j| / m \tag{8}$$

$$\text{MSE} = \sum_{j=1}^{m} (y_j - \hat{y}_j)^2 / m \tag{9}$$

where $y_i$ represents the actual values of the calculation sample, $\hat{y}_i$ represents the corresponding calculated values, $\overline{y}_i$ represents the mean of the calculation sample, and n is the size of the calculation sample. $y_j$ represents the actual values of the test sample, $\hat{y}_j$ represents the corresponding predicted values, and m is the size of the test sample.

The *p*-value is used to test the significance of differences between the predicted results $\hat{y}_j$ and the actual values $y_j$, indicating the reliability of the model. Calculation is performed using paired *t*-tests [27], and the significance criterion is set at $p < 0.05$.

*3.5. Calculation of Future Maintenance Quantity*

The projected number of buildings of a certain category within the jurisdiction in the coming year is calculated using the following equation:

$$\text{Projected number of a category} = \text{projected category proportion} \times \text{number of samples in the jurisdiction} \tag{10}$$

## 4. Results

### 4.1. Results of Data Standardization

Based on the original dataset and Equation (1), the proportions of the third category buildings in each jurisdiction's sample set are calculated separately. To eliminate the interference of temporarily unassessed buildings on the overall proportion, only the sum of the first–third-category buildings is calculated. The percentages are then converted to decimals. The jurisdiction sample sets are uniformly referred to by the abbreviations of the governing units, and the results are shown in Table 3. The data for 2022 in the table are specifically used for model testing.

**Table 3.** The standardization sample set of the third category buildings in each jurisdiction along the Yangtze River trunk waterway.

|  | YBB | LZB | CQB | YCB | JZD | YYD | WHB | JJD | WHD | ZJD | SHD |
|---|---|---|---|---|---|---|---|---|---|---|---|
| Proportion in 2017 |  |  |  |  |  |  |  | 0.022 | 0.000 |  | 0.000 |
| Proportion in 2018 | 0.125 | 0.095 | 0.143 | 0.000 | 0.026 | 0.102 | 0.042 | 0.044 | 0.040 |  | 0.000 |
| Proportion in 2019 | 0.083 | 0.070 | 0.077 | 0.000 | 0.026 | 0.037 | 0.024 | 0.059 | 0.016 |  | 0.000 |
| Proportion in 2020 | 0.083 | 0.091 | 0.071 | 0.031 | 0.026 | 0.019 | 0.043 | 0.026 | 0.013 | 0.082 | 0.000 |
| Proportion in 2021 | 0.077 | 0.068 | 0.200 | 0.031 | 0.000 | 0.036 | 0.043 | 0.000 | 0.000 | 0.102 | 0.000 |
| Proportion in 2022 | 0.040 | 0.044 | 0.182 | 0.000 | 0.000 | 0.070 | 0.022 | 0.000 | 0.000 | 0.102 | 0.000 |
| Total quantity in 2022 | 25 | 45 | 11 | 31 | 77 | 57 | 45 | 38 | 68 | 49 | 41 |

Notes: Blank cells indicate the lack of basic information for the year and jurisdiction to calculate category proportion. The last row lists the total quantities of the first to third category buildings within the jurisdiction in 2022.

### 4.2. Model Calculation and Test Results

Firstly, the data for each jurisdiction from 2017 to 2021 in Table 3 are input into Formulas (2)–(5) to calculate the model coefficients for JMA, LR, and QCR. Then, Formulas (6) and (7) are used to calculate the model error parameters RMSE and $R^2$. Finally, the JMA, LR, and QCR models are utilized to predict the maintenance proportion values for buildings in each jurisdiction in 2022. The results are all listed in Tables 4–6. Please note that historical data for Zhenjiang consist of only two points, and historical data for Shanghai are all 0, making it impossible to fit them with QCR. Particularly, due to the impossibility of negative values for the proportion and quantity of building categories, any calculated negative values should automatically be set to zero.

**Table 4.** JMA model coefficients, error parameters, and predicted proportion (2022).

| Jurisdiction | Historical Mean | RMSE | Predicted Proportion | 95% Confidence Interval |
|---|---|---|---|---|
| YBB | 0.092 | 0.022 | 0.092 | $0.092 \pm 0.035$ |
| LZB | 0.081 | 0.014 | 0.081 | $0.081 \pm 0.022$ |
| CQB | 0.123 | 0.061 | 0.123 | $0.123 \pm 0.097$ |
| YCB | 0.016 | 0.018 | 0.016 | $0.016 \pm 0.029$ |
| JZD | 0.019 | 0.013 | 0.019 | $0.019 \pm 0.021$ |
| YYD | 0.049 | 0.037 | 0.049 | $0.049 \pm 0.058$ |
| WHB | 0.038 | 0.009 | 0.038 | $0.038 \pm 0.015$ |
| JJD | 0.030 | 0.022 | 0.030 | $0.030 \pm 0.028$ |
| WHD | 0.014 | 0.016 | 0.014 | $0.014 \pm 0.020$ |
| ZJD | 0.092 | 0.014 | 0.092 | $0.092 \pm 0.130$ |
| SHD | 0.000 | 0.000 | 0.000 | $0.000 \pm 0.000$ |

**Table 5.** LR model coefficients, error parameters, and predicted proportion (2022).

| Jurisdiction | $\alpha$ | $\beta$ | RMSE | $R^2$ | Predicted Proportion |
|---|---|---|---|---|---|
| YBB | −0.014 | 29.220 | 0.010 | 0.709 | 0.056 |
| LZB | −0.006 | 12.204 | 0.010 | 0.305 | 0.066 |
| CQB | 0.017 | −33.388 | 0.049 | 0.124 | 0.164 |
| YCB | 0.013 | −25.228 | 0.007 | 0.800 | 0.047 |
| JZD | −0.008 | 15.896 | 0.007 | 0.610 | 0.000 |
| YYD | −0.022 | 43.508 | 0.021 | 0.576 | −0.005 |
| WHB | 0.002 | −4.914 | 0.008 | 0.116 | 0.044 |
| JJD | −0.006 | 12.469 | 0.018 | 0.188 | 0.012 |
| WHD | −0.003 | 5.468 | 0.014 | 0.068 | 0.006 |
| ZJD | 0.020 | −41.143 | 0.000 | 1.000 | 0.122 |
| SHD | 0.000 | 0.000 | 0.000 | | 0.000 |

Note: Historical data for SHD are all 0, making it impossible to calculate $R^2$.

**Table 6.** QCR model coefficients, error parameters, and predicted proportion (2022).

| Jurisdiction | a | b | c | RMSE | $R^2$ | Predicted Proportion |
|---|---|---|---|---|---|---|
| YBB | 0.009 | −35.615 | 35,976.470 | 0.005 | 0.921 | 0.100 |
| LZB | 0.001 | −2.776 | 2809.339 | 0.010 | 0.308 | 0.069 |
| CQB | 0.049 | −196.385 | 198,283.400 | 0.008 | 0.976 | 0.407 |
| YCB | 0.000 | 0.013 | −25.228 | 0.007 | 0.800 | 0.047 |
| JZD | −0.006 | 25.538 | −25,779.100 | 0.003 | 0.926 | −0.032 |
| YYD | 0.021 | −83.325 | 84,159.560 | 0.001 | 0.998 | 0.098 |
| WHB | 0.004 | −17.442 | 17,610.040 | 0.006 | 0.405 | 0.066 |
| JJD | −0.010 | 41.796 | −42,186.300 | 0.005 | 0.930 | −0.061 |
| WHD | −0.006 | 24.437 | −24,666.100 | 0.010 | 0.546 | −0.037 |

### 4.3. Comprehensive Evaluation of the Model

To comprehensively evaluate the three models, their error parameters are calculated. First, the average values of RMSE and $R^2$ from Tables 4–6 are calculated. Next, the predicted proportions (Tables 4–6) and the actual proportions (Table 3) for 2022 are input into Formulas (8) and (9) to calculate MAE and MSE. Finally, paired *t*-tests are conducted on the test results to obtain *p*-values. All results are listed in Table 7.

**Table 7.** Summary of error parameters for the JMA, LR, and QCR models.

| Model | $(RMSE)^-$ | $(R^2)^-$ | MAE | MSE | *p*-Value |
|---|---|---|---|---|---|
| JMA | 0.021 | | 0.025 | 0.001 | 0.658 |
| LR | 0.013 | 0.450 | 0.022 | 0.001 | 0.740 |
| QCR | 0.006 | 0.757 | 0.062 | 0.007 | 0.372 |

Note: The symbol $^-$ represents the statistical average of the error parameters. The $R^2$ of the JMA model cannot be calculated.

Results from the analysis of RMSE and $R^2$ reflect the goodness of fit for the models, with smaller values indicating a better fit. Table 7 proves that the models' goodness-of-fit ranks as follows: QCR > LR > JMA. The MAE and MSE from the test results directly indicate the predictive accuracy and generalization ability of the models, with smaller values indicating more accurate predictions. The results show that the errors of the JMA and LR models are quite close, with LR having slightly smaller errors, while the QCR model has significantly larger errors. The *t*-test results indicate that the *p*-values of all three models are greater than 0.05, suggesting that the differences are not statistically significant. This implies that the differences between predicted and actual values may be due to random factors rather than model issues. Notably, the p-values for the JMA and LR models are much higher than those of the QCR model, indicating relatively reliable predictive results.

The larger RMSE and lower $R^2$ in the models are likely due to the simplicity of the models and the small sample size, which can lead to larger random errors. However,

this should not directly discredit the usability of the models. The more complex models may have improved the goodness of fit but could lead to overfitting and reduced model generalization ability. For practical engineering applications, ease of operation, accuracy of the final results, and generalization ability are clearly more important than goodness of fit.

Overall, although QCR shows higher goodness of fit, the results exhibit significant errors, indicating lower predictive accuracy and generalization ability. Therefore, QCR is not recommended. On the other hand, JMA and LR, despite having lower goodness of fit, demonstrate fewer errors in the results, suggesting higher predictive accuracy and generalization ability. As a result, JMA and LR are recommended as the final predictive models. The difference in predictive accuracy between the two models is not substantial, and the choice between them can be based on practical considerations in engineering applications.

*4.4. The Final Result of the Number of Future Repairs in Each Jurisdiction*

Combining the total counts for 2022 (Table 3), the predicted proportions for the JMA and LR models (Tables 4 and 5) are input into Equation (10) to calculate the final maintenance quantities for the waterway improvement buildings in various jurisdictions along the Yangtze River trunk waterway for 2022 (Figure 6).

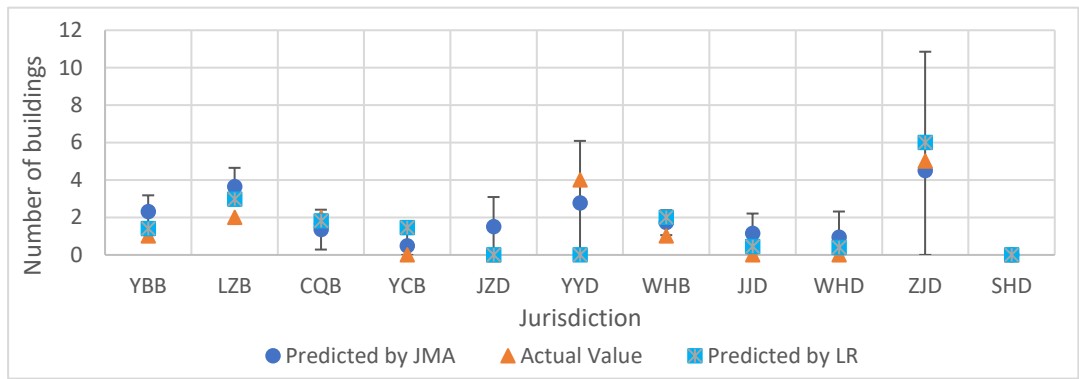

**Figure 6.** The final predicted quantity and the actual measured quantity of improvement buildings in need of maintenance in each jurisdiction for 2022. (The error bars in the figure represent the confidence interval of the JMA model, and in cases where the calculated quantity of buildings is negative, the values have been automatically adjusted to zero.)

Figure 6 displays the final results predicted by the models and the actual maintenance quantities for 2022. Except for LZB, the actual counts are within the confidence intervals of the JMA predictions. The large error bar range for ZJD is related to the small size of the time-series sample set, which contains only two data points. The small sample size leads to a small freedom degree for the t-distribution, resulting in a large critical value (12.706) and, hence, a large boundary for the confidence interval. Thus, a small sample size can reduce the predictive accuracy of the model, and it is advisable to have a sample size of at least four or more for better results.

The final results suggest that except for the significant difference between the LR model's prediction and the actual values in YYD, the predictions for other jurisdictions are relatively close to the actual values. The JMA model provides accurate predictions and has confidence intervals encompassing almost all the actual measurements, thus increasing the credibility of the final results. Therefore, the JMA model's predictive performance appears to be ideal.

## 5. Comparison and Discussion

*5.1. The Advantages, Disadvantages, and Applicability of the Proposed Technology*

The existing maintenance decision-making process is formulated based on assessing the technical conditions of waterway improvement buildings for the current year. In contrast, the network-level assessment framework proposed in this study involves forecasting

maintenance quantities for the future year in advance. Subsequently, maintenance funds for different regions are allocated ahead of time based on these predictions, optimizing the maintenance decision-making process. Therefore, this proposed framework is more forward-thinking and efficient than the existing approach. The Network-level Evaluation Prediction Model (NEP) proposed in this paper was compared with the efficiency of known existing technologies, and each characteristic was summarized and listed in Table 8. The existing technologies included the methods used in the Technical Code of Inland Waterway Maintenance [24] (TCIWM), the AHP-Improved CRITIC Combination Weighting Optimization Model [14] (AHP-ICRITIC), the Fuzzy Bayesian Network Model [16] (FBN), and the SVM Predicting Model [11] (SVMP).

**Table 8.** Comparison of characteristics of proposed in this paper and known existing technologies.

| Features | NEP | TCIWM | AHP-ICRITIC | FBN | SVMP |
|---|---|---|---|---|---|
| Multicriteria Attribute | Comprehensive Qualitative | Comprehensive Qualitative | Comprehensive Quantitative | Comprehensive Quantitative | Single Quantitative |
| Characteristic of Indicator Weight | Historical Assessment Data Based | Subjective Experience Based | Subjective and Objective Integration Based | Trial Calculation Based | |
| Time Status of Evaluation Objects | Future | Present | Present | Present | Future |
| Number of Evaluation Objects | Overall Buildings in the Jurisdiction Area | Single Building | Single Building | Single Building | Riverbank Building Group |
| Sample Size Requirement | Low | Low | High | Low | High |
| The Method's Understandability and Operability | Simple and Easy to Understand | Operational Complexity | Rather Complicated | Rather Complicated | Rather Complicated |
| The Method's Reliability | Theoretically Reliable | Practically Reliable | Theoretically Reliable | Theoretically Reliable | Theoretically Reliable |
| Maintenance Decision-making Timing | Ahead of Time | Lagging | Lagging | Lagging | Ahead of Time |

From the table, it can be intuitively seen that the proposed technology has a higher overall efficiency compared to the known existing technologies. This means that this technology can more accurately predict the maintenance needs of buildings, thereby improving the efficiency of remedial work.

The network-level assessment predictive model proposed in this paper is straightforward, easy to understand, and does not require a large sample. It has demonstrated sufficient reliability, making it highly suitable for wide application in practical engineering. The model is particularly well-suited for small sample predictions. However, the prediction accuracy is still related to the size of the time-series sample. In real-world applications, new evaluation data can be continuously added to the sample set, allowing for ongoing refinement of model parameters and improving prediction accuracy over time.

The premise for applying this model is based on the assumption that the waterway conditions within the jurisdictional areas remain relatively stable with minimal changes over the preceding years. Therefore, this model is well-suited for predicting waterway improvement building maintenance status of the upcoming year and less suitable for scenarios with a significantly longer period. Similarly, the model does not apply to situations with a sudden and dramatic change in waterway conditions within the jurisdictional areas. Such changes may result from events like the construction of dams upstream, landslides affecting flow rates, flow velocities, water levels, riverbed alterations, or severe structural damage and deformations of buildings due to human activities. In such cases, a reevaluation of the structural stability of buildings, risk factors, and the extent of changes in waterway elements should be conducted through further simulation experiments or theoretical analyses.

Additionally, it is essential to note that the predictive accuracy of the proposed model is limited. Therefore, it should be utilized as a reference for optimizing regional waterway

infrastructure maintenance strategies but should not be applied for individual building maintenance assessments. In practical applications, a comprehensive maintenance decision-making and assessment process should consider not only the model's predictive outcomes but also various other relevant factors.

*5.2. Further Discussion on the Research Results*

Based on the final results presented in this paper, the JMA model demonstrates the best predictive performance. Conducting network-level assessments of improvement buildings on a jurisdictional basis is reasonable and aligns with practical management needs. The research results can provide a scientific basis for setting maintenance budgets for waterway improvement buildings and offer ideas for inland waterway management and maintenance. The proposed prediction method applies to not only the Yangtze River trunk waterway but also other inland waterways, which is of great practical significance.

The theoretical contribution of this study mainly revolves around the improvement of the network-level evaluation framework for optimizing maintenance decisions of inland waterway infrastructure. This is an important extension and new application, as most previous research was limited to road traffic and other related fields. As a result, this study enriches the literature on predictive maintenance models for infrastructure and paves the way for future research, promoting a theoretical understanding of water transport infrastructure management.

From a management perspective, this study provides insights into the optimal allocation of limited maintenance funds through predictive models. The proposed model provides a tool for a quick evaluation of overall risk, which may significantly improve management decision making and infrastructure maintenance practices. By comparing different models, this study also offers clear guidelines on the choice of predictive models, where the jurisdiction moving average (JMA) model was identified as the most ideal due to its larger confidence interval.

At the policy level, this study provides important insights for the improvement, management, and budgetary decision making for waterway infrastructures. By applying network-level evaluation, this study recommends a resource allocation approach that considers the condition of inland waterway construction technology. Decision makers can use the insights from this study to develop sound and efficient infrastructure maintenance policies, optimize budget allocation, and ultimately promote the sustainable development of the water transport industry.

**6. Conclusions**

This study applies the concept of network-level evaluation innovatively to waterway transportation infrastructure and proposes an optimal maintenance predictive model. The results show that, although the QCR model has a high fitting degree, its accuracy and generalization ability are not high, and therefore it is not recommended. In contrast, the MAJ and LR models have higher prediction accuracy and generalization ability, especially the MAJ model, which has the best performance. Using historical inspection data to predict the future risk status of canal remediation buildings is reasonable and feasible. This provides a new research idea for the management of the Yangtze River main channel and also lays the foundation for further research in network-level assessment of waterway transportation infrastructure. In the future, more detailed data will be collected to enhance predictive accuracy. Additionally, incorporating soft computing techniques such as Markov models, random fields, rough sets, and neural networks into network-level assessments will enable condition predictions over longer periods. Furthermore, this research will expand to the minimization of overall maintenance costs and the maximization of network performance indicators, which will involve ongoing improvements to the technical framework and predictive models for network-level assessments of waterway improvement buildings.

**Author Contributions:** Conceptualization, F.Z.; Methodology, J.S.; Software, J.S.; Validation, J.L.; Formal analysis, F.Z. and J.L.; Investigation, H.L.; Resources, H.L.; Data curation, B.Z.; Writing—original draft, F.Z.; Writing—review & editing, H.L.; Visualization, B.Z.; Supervision, P.W.; Project administration, P.W.; Funding acquisition, P.W. All authors have read and agreed to the published version of the manuscript.

**Funding:** This research was funded by the "National Key Research and Development Program of China" (grant number 2018YFB1600403), "Natural Science Foundation of Chongqing, China" (grant number cstc2021jcyj-msxmX0667), "Key Laboratory of Hydraulic and Waterway Engineering of the Ministry of Education, Chongqing Jiaotong University, China" (grant number SLK2023B09), and "Research and Innovation Program for Graduate Students in Chongqing, China" (grant number CYB21217).

**Institutional Review Board Statement:** Not applicable.

**Informed Consent Statement:** Not applicable.

**Data Availability Statement:** Data are contained within the article.

**Acknowledgments:** The authors would like to thank the Changjiang Waterway Bureau of China for providing the required data for this article. We also thank Yang Chengyu, Yu Tao, and Han Linfeng of Chongqing Jiaotong University for their guidance on this article.

**Conflicts of Interest:** The authors declare no conflict of interest.

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
