# Peer review of "Inland Waterway Infrastructure Maintenance Prediction Model Based on Network-Level Assessment"

_sustainability, doi:10.3390/su152216027_

Round 1

Reviewer 1 Report (Previous Reviewer 1)

Comments and Suggestions for Authors

sustainability-2693475-peer-review-v1
Research on Maintenance Prediction Model for Inland Waterway Improvement Buildings Based on Network-Level Assessment

In this paper, author shares a study on research on maintenance prediction model for inland waterway improvement buildings based on network-level assessment. Although the subject is interesting, I cannot accept the paper in the current form due to the following reasons:

1.               The title of the manuscript should be trimmed and make it more appealing to the readers.

2.               Lines 12-29: Generally, the abstract of paper is based on research aim/purpose, research method, and key findings. Abstract of this paper is well written but it is required to highlight the key findings of the study.

3.               Lines 35-42: In introduction, before starting the mentioned references, there is a need to add 8-9 lines related to the subject of the paper and write in general introduction. After that you should connect them with the references.

4.               Lines 35-120: The major defect of this study is the debate or Argument is not clearly stated in the introduction session. Hence, the contribution is weak in this manuscript. I would suggest the author enhance your theoretical discussion and arrives at your debate or argument.

5.               Novelty not clear. What are the major issues in the existing methods? Author must explain. As a reader, I am interested to see the research questions(s) to be presented in the Introduction.

6.               Motivation behind the work is missing. After discussing relevant literature, authors should clearly say what motivated them to propose this work.

7.               Structure your Introduction based on three paragraphs: (1) The question your article seeks to address, (2) The ‘state-of-the-art’ on this particular issue, and (3) What your paper is going to actually do (as this is the bridge into the Methods).

8.               The manuscript is poorly structured. Why is a literature review jam-packed within the introduction and not presented as a standalone section?

9.               Do the authors understand the reason for peer review? The reviews process aims to ensure the proposed methods are replicable or reproducible. The expert opinion also is meant to help the authors identify the weakness in their approach, experiment, and manuscript's content. Thus, a peer review is not just meant for questions and answers. Suppose the author wants to resubmit the same manuscript for re-review in this journal or other journals. In that case, the authors should refer to papers published in top-ranked journals to see how the contents are presented and organized well as the quality. The authors should also discuss relevant recent works to give the reader a thorough grounding on the state-of-the-art.

10.           Furthermore, the analysis of the results presented in the manuscript requires more compelling and comprehensive treatment. It is crucial to delve deeper into the implications of the results, highlight their significance, and discuss their broader implications in the context of the relevant field. A more extensive exploration of the outcomes would enable a more thorough understanding of the research findings and contribute to the scholarly discourse.

11.           The comparative part of this paper is very weak. Author should make more comparative study to compare the efficiency of the proposed techniques with well-known existing techniques.

12.           The authors should try to improve the quality of Figures 1 & 2. It cannot be tolerated.

13.           May I suggest that the authors try to introduce a separate section altogether for presenting the Theoretical Implications, Managerial insights, and Policy implications of this study? Currently, they are absent from the manuscript.

14.           Lines 380-406: Your conclusion is like an Introduction Section. Conclusion is very long/lengthy. In the conclusion section, please revise it and improve it by re-organizing it into one paragraph only including the suggested future work.

***

Author Response

Reviewer 2 Report (Previous Reviewer 3)

Comments and Suggestions for Authors

Dear Authors, thank you for adapting the new text of the article to my recommendations basing on the previous manuscript. I strongly believe that the improvements introduced have a big impact on the transparency and logic of the presented research material. I wish you success in further research.

Regards,

Reviewer

Author Response

Dear reviewer, Thank you very much for your valuable comments and suggestions. Your constructive feedback has been truly instrumental in enhancing the quality and clarity of our manuscript. We were highly motivated by the encouraging words you've provided regarding the revisions we have made. It has indeed been our endeavor to ensure that the research material presented is both transparent and logical in its approach. We are grateful for your good wishes and we hope to continue to present our research with the same diligence and commitment. We sincerely appreciate your time and effort in helping us improve our work, and we are looking forward to our future interactions. Thank you once again for your support and guidance. Best regards, Fan Zhang, Pingyi Wang, Huaihan Liu, Bin Zhang, Jianle Sun and Jian Li zff@yznu.edu.cn

Reviewer 3 Report (Previous Reviewer 2)

Comments and Suggestions for Authors

In this paper, the spatiotemporal distribution characteristics of improvement building service status were analyzed. The topic is interesting. However, the paper needs major revisions to improve its quality.

1. The novelty of this paper should be further clarified.

2. The limitation of this paper should be figured out.

3. What is the main question addressed by the research?

4. Can the method proposed in this paper be used in the field, or can it be used in other fields as well?

5. The contribution should be further clarified.

6. The proposed prediction method in this paper should be father validated.

7. The Jurisdiction moving average analysis is unclear. What’s the moving window size?

8. Linear regression and Quadratic curve regression mean the data has linear and nonlinear relationship, respectively. The relationship between data needs to be illustrated through scatter plots firstly, and then the suitable model can be chosen.

9. The sample data only covers 2017 to 2021. There are only five points. The small sample size will make the results unconvinced.

10. What does “Network-Level” mean? What’s the differences between network-level and non-network-level predictions?

11. It should be revised to no more than 7 keywords through deleting the unnecessary ones.

Comments on the Quality of English Language

Moderate editing of English language required

Round 2

Reviewer 1 Report (Previous Reviewer 1)

Comments and Suggestions for Authors

sustainability-2693475-peer-review-v2

Research on Maintenance Prediction Model for Inland Waterway
Improvement Buildings Based on Network-Level Assessment.

After a careful evaluation of the revised manuscript, it is difficult to recommend the very manuscript for publication in its entirety. The revised paper has not been improved, and I think the authors' attitude to the manuscript is rather casual. The authors' responses to the referee's comments are incomplete, confusing, and incomprehensible. I do not see anything new such as the appropriateness, novelty and general significance. There are also several technical content and quality issues. I regret to recommend to reject this paper.

Comments on the Quality of English Language

Please see the report

Reviewer 3 Report (Previous Reviewer 2)

Comments and Suggestions for Authors

The English should be improved.  There are some typo poblems.

Comments on the Quality of English Language

The English should be improved.  There are some typo poblems.

This manuscript is a resubmission of an earlier submission. The following is a list of the peer review reports and author responses from that submission.

Round 1

Reviewer 1 Report

Comments and Suggestions for Authors

sustainability-2550882-peer-review-v1
Spatiotemporal analysis and prediction of Yangtze River trunk waterway improvement building technical condition categorization.

In this paper, author shares a study on spatio-temporal analysis and prediction of Yangtze river trunk waterway improvement building technical condition categorization. The topic seems to be interesting. However, the quality of proposed work is not good according to the following comments:

1.      Lines 30-72: In introduction section authors do not explain the significance of proposed method.
2.      There is a major need to revise the article, because there are several grammatical mistakes in native English writing.
3.      Overall presentation of paper is not attractive.
4.      Page 10-12: The advantages and disadvantages section is not included.
5.      Lines 30-72: Novelty of new concept is not enough for publication.

6.      I feel that although this may be a useful exercise, the paper does not come up to the international standards for publication. There is hardly any original contribution in this paper and it adds very little to the published literature.

***

Comments on the Quality of English Language

There is a major need to revise the article, because there are several grammatical mistakes in native English writing.

Reviewer 2 Report

Comments and Suggestions for Authors

In this paper, the spatiotemporal distribution characteristics of improvement building service status were analyzed, and a novel method of predicting future technical condition categories in each jurisdiction was proposed, which can provide a scientific basis for setting waterway improvement building maintenance budgets. However, the paper needs some revisions to improve its quality.

1. "expectation and quadratic regression " is not given in the key words, but it appear many times in the article.

2. The novelty of this paper should be further clarified.

3. The number of decimal places in the data in the chart should be consistent.

4. In Table 3, the sum of category statistical frequencies of the same year in the sample set is not all 1.00.

5. In Table 4,the number of buildings should be rounded up to integers.

6. In line 244 of the article "The sample size", The first letter is incorrectly capitalized.

7. The limitation of this paper should be figured out.

1.   What is the main question addressed by the research?

2.   Can the method proposed in this paper be used in the field, or can it be used in other fields as well?

3.   What is its contribution to the subject area compared with other published material?

4.   Is the prediction method proposed in this paper reliable? What are the limitations?

5.   Are the references appropriate?

Please include any additional comments on the formulas, figures and tables.

Comments on the Quality of English Language

Extensive editing of English language required

Reviewer 3 Report

Comments and Suggestions for Authors

Dear Authors, I would like to thank you for the opportunity to read your manuscript entitled “Spatiotemporal analysis and prediction of Yangtze River trunk waterway improvement building technical condition categorization”.

The overall manuscript is well presented with minor spelling or grammar mistakes. The introductory part is based on a references list comprising only 19 positions with self-citations, issued in general in between 2014 and 2022. The references list does not provide sufficient support for determining the actual status of the research in the field.

The general work is very interesting, as the revision of spatiotemporal factors influencing the improvement of Yangtze River is important for its development.  

Here are some issues concerning your paper:

1.      The entire text of the article should be corrected in terms of editing due to the variety of font used and citations that do not meet the requirements of the journal.

2.      The overall scientific purpose of the article should be stated clearly in the introduction and underlined in the abstract despite the fact that it results from the presented text.

3.      References should be numbered in order of appearance.

4.      The Literature Review part is practically missing and only shown in the Introduction. The current state of the research is not shown and cited. The knowledge gap is unclear. There should be an explanation of the previous work done not only by the authors but other researchers in these field. It is well presented with the examples from other countries when explaining the structure of the empirical research.

5.      The Methodology part is missing in the manuscript. It only presents results.

6.      The subject of research, which is the Yangtze River, together with the districts presented in Table 1, should be depicted on the map. It is worth to show the upper, medium and lower parts on it.

7.      In part 1.1 please explain what are the “regulated buildings” or “improvement buildings”. Is it the same? What are their types? What is the need for their classification?

8.      All Figures and Tables are very well presented and readable with appropriate citations in the main text.

9.      The Discussion on the obtained research results should be reconsidered. There is no information on the current new results and an attitude to the existing research, which could emphasize the importance of the work done.

10.   The Conclusion on the obtained research results should be reconsidered. It is too short and laconic. Future research directions and the significance of the results of the research achieved are underlined and explained in conclusion part. It should also be edited because it is too short and laconic.

11.   References must be reconsidered. The list is too short and do not present what already has been done in the field.

Reviewer

Comments on the Quality of English Language

English language fine.